# Preparation and Evaluation of Inhalable Microparticles with Improved Aerodynamic Performance and Dispersibility Using L-Leucine and Hot-Melt Extrusion

**DOI:** 10.3390/pharmaceutics16060784

**Published:** 2024-06-08

**Authors:** Jin-Hyuk Jeong, Ji-Su Kim, Yu-Rim Choi, Dae Hwan Shin, Ji-Hyun Kang, Dong-Wook Kim, Yun-Sang Park, Chun-Woong Park

**Affiliations:** 1Department of Pharmacy, Chungbuk National University, Cheongju 28644, Republic of Korea; jinddong92@gmail.com (J.-H.J.); 96kimjs@gmail.com (J.-S.K.); choiyurim0730@gmail.com (Y.-R.C.); dshin@chungbuk.ac.kr (D.H.S.); jhkanga@jbnu.ac.kr (J.-H.K.); 2Institute of New Drug Development and Respiratory Drug Development Research Institute, School of Pharmacy, Jeonbuk National University, Jeonju 54896, Republic of Korea; 3College of Pharmacy, Wonkwang University, Iksan 54538, Republic of Korea; pharmengin1@wku.ac.kr; 4Research & Development Center, P2K Bio, Cheongju 28160, Republic of Korea; bbabbak22@naver.com

**Keywords:** dry-powder inhaler, hot-melt extrusion, L-leucine, itraconazole, aerodynamic performance

## Abstract

Dry-powder inhalers (DPIs) are valued for their stability but formulating them is challenging due to powder aggregation and limited flowability, which affects drug delivery and uniformity. In this study, the incorporation of L-leucine (LEU) into hot-melt extrusion (HME) was proposed to enhance dispersibility while simultaneously maintaining the high aerodynamic performance of inhalable microparticles. This study explored using LEU in HME to improve dispersibility and maintain the high aerodynamic performance of inhalable microparticles. Formulations with crystalline itraconazole (ITZ) and LEU were made via co-jet milling and HME followed by jet milling. The LEU ratio varied, comparing solubility, homogenization, and aerodynamic performance enhancements. In HME, ITZ solubility increased, and crystallinity decreased. Higher LEU ratios in HME formulations reduced the contact angle, enhancing mass median aerodynamic diameter (MMAD) size and aerodynamic performance synergistically. Achieving a maximum extra fine particle fraction of 33.68 ± 1.31% enabled stable deep lung delivery. This study shows that HME combined with LEU effectively produces inhalable particles, which is promising for improved drug dispersion and delivery.

## 1. Introduction

Pulmonary drug delivery is a promising route for the administration of biologically unavailable compounds, particularly those limited by first-pass metabolism or poor gastrointestinal absorption [1,2]. Itraconazole (ITZ), a triazole antifungal agent, is a pivotal medication for treating allergic bronchopulmonary aspergillosis [3]. However, ITZ is a weakly basic drug (pKa = 3.7) that ionizes effectively only at a low pH, exhibits minimal solubility and high permeability, and is classified under the Biopharmaceutics classification system class II [4]. Its low solubility and dissolution rates limit its oral bioavailability, which is approximately 55%. Current treatments are administered orally and intravenously. However, these conventional administration routes require high systemic concentrations to achieve effective lung levels, potentially causing severe side effects such as hepatotoxicity, congestive heart failure, and metabolic interactions [5,6]. Furthermore, the inability to deliver optimal amounts of the drug to the infection site can lead to treatment failure or resistance. The inhalation of antifungals is becoming increasingly important in addressing issues associated with conventional treatment methods [7].

Among pulmonary drug delivery formulations, dry-powder inhalers (DPIs) are the most traditional and continue to be extensively developed [8]. They offer chemical stability and do not require propellants, unlike liquid formulations [9], with a lower risk of issues arising from the formulation [10]. They are generally suitable for delivering high drug concentrations and ensure reproducibility and reliability [11,12]. To reach deeper lung penetration, it is essential for the drug particles in DPIs to have aerodynamic diameters ranging from 0.5 μm to 5 μm [13]. Unfortunately, these fine powders often exhibit high cohesiveness, leading to poor flowability [14]. To address this issue, the most commonly adopted approach is to create a homogeneous interactive mixture of fine drug particles and carrier particles [15,16]. However, controlling triboelectric forces is difficult and complex, making it a major challenge in DPI formulation to ensure the homogeneity and stability of the mixture during powder handling and administration [17,18].

Hot-melt extrusion (HME) is a process in which materials are pumped through a die into a uniform shape by rotating screws at high temperatures. In pharmaceuticals, HME presents several clear benefits, including excellent homogeneity, simple processing steps, continuous manufacturing, low cost, and scalability for industrial production [19,20]. In particular, the HME process involves intense mixing and agitation, resulting in excellent content uniformity of extruded products and more uniform dispersion of drug particles for post-processing drug content [21,22,23,24]. However, particles prepared via HME possess aerodynamic diameters that are too large for inhalation [25]. Therefore, we introduced jet milling to reduce the particle size of the extrudates and achieve an ideal aerodynamic behavior. In preliminary studies, post-extrusion jet-milled microparticles were shown to maintain drug crystallinity while reducing crystallinity. The reduced crystallinity presents two advantages over simple amorphous solid dispersions. First, it offers superior physical stability and does not cause the rapid recrystallization issues inherent in amorphous formulations [26]. Moreover, it has significantly lower hygroscopicity than amorphous materials, aiding in maintaining satisfactory aerodynamic characteristics over its shelf life [23]. Second, it does not require specific drug-excipient interactions, making it applicable to a wide range of drugs [20]. For these reasons, recent studies have introduced HME in the development of dry-powder inhalers to reduce the crystallinity of drugs, thereby improving solubility and physical stability [23,27]. Nevertheless, the need to enhance aerodynamic performance due to increased moisture affinity from improved solubility remains a challenge to be addressed.

Amino acids have been extensively studied as excipients in inhalable dry powders. In particular, L-leucine (LEU), a nonpolar aliphatic amino acid, is of particular interest because of its unique properties [28]. Generally, LEU is added as an excipient to spray-dried inhalable dry powders to enhance dispersibility, potentially improve aerosol performance, and provide moisture protection, thereby increasing stability [29]. In spray drying (SD), LEU concentrates on the surface of the spray droplets, imparting hydrophobicity to the particle exterior and potentially altering the particle morphology [30,31]. Compared to other amino acids, the effectiveness of LEU is attributed to its hydrophobic nature and nonpolar side chain. Prior to the drying stage, its surfactant properties reduce the surface tension of the aqueous feed, decreasing the size of droplets formed during spraying and promoting the formation of smaller particles. Additionally, when more than one compound is present in the solution, the surface of the sprayed droplets is populated by leucine molecules, with their hydrophobic side chains oriented toward the air and their hydrophilic parts facing the water inside the droplets [32]. As water continues to evaporate, the concentration of leucine at the droplet surface increases until it reaches a supersaturated state, leading to crystallization. It is particularly noted that in leucine droplets with an initial diameter of 50 µm, a leucine shell can be detected when the supersaturation ratio reaches 5. However, the use of LEU in DPIs has only been explored using SD [30]. In this study, we hypothesized that the introduction of LEU into the HME process would maintain the dispersiveness enhancement and moisture protection effects expected from its hydrophobic surface in SD.

The objectives of this study were to prepare inhalable microparticles using HME and jet milling with the introduction of LEU, verify the anticipated effects of LEU, and develop a high-quality DPI formulation. ITZ, with its high hydrophobicity, and high thermal stability, is suitable for HME [4,33]. The hot-melt-extruded DPI formulation with LEU was expected to provide (i) superior physicochemical homogeneity and (ii) excellent aerodynamic performance and appropriate deposition sites.

## 2. Materials and Methods

### 2.1. Materials

Itraconazole (ITZ) was purchased from SMS Pharmaceuticals Ltd. (Hyderabad, India). D-Mannitol (MAN) was purchased from Sigma-Aldrich Co., Ltd. (Darmstadt, Germany). LEU was purchased from Sigma-Aldrich Co., Ltd. (Darmstadt, Germany). The water was purified by filtration in the laboratory. High-performance liquid chromatography-grade (HPLC)-grade solvents were used for the analysis. HPLC-grade ethanol (EtOH) and acetonitrile (ACN) were purchased from Honeywell Burdick and Jackson (Charlotte, NC, USA). All reagents were of analytical grade and used without further purification.

### 2.2. Preparation of ITZ Microparticles

#### 2.2.1. Preparation of ITZ Microparticles Using Co-Jet Milling

The ratio of raw ITZ to MAN was fixed at 2:8 (*w/w*), with LEU mixed at a ratio of 0%, 0.1%, 1.0%, and 10.0% (Table 1). Before milling, the mixture was mixed physically for 30 min using a T2F Turbula^®^ shaker-mixer (WAB-GROUP^®^, Muttenz, Switzerland). These physical mixtures (PMs) are used as the PM formulations. The physical mixtures (PMs) were co-milled using an air jet mill (A-O JET MILL, J S Tech Co., Ltd., Sacheon, Republic of Korea) with the following parameters: G nozzle (MPa), 0.60; P nozzle (MPa): 0.65 (Figure 1). These co-jet milled (JM) preparations were used as the JM formulations. The JM ITZ microparticles were collected in glass vials and stored in a desiccator with silica beads until further use.

#### 2.2.2. Preparation of ITZ Microparticles Using HME and Jet Milling 

The ratio of raw ITZ to MAN was fixed at 2:8 (*w/w*), and LEU was mixed at a ratio of 0%, 0.1%, 1.0%, and 10.0% (Table 1). Before melt extrusion, the mixtures were physically premixed for 30 min using a T2F Turbula^®^ shaker-mixer. The PMs were manually fed into a co-rotating twin-screw extruder (Process 11; Thermo Electron GmbH, Karlsruhe, Germany). Extrusion was performed at a screw speed of 150 rpm and a temperature setting of 160 °C across all eight zones [23]. These temperatures were set at approximately 15 °C lower than the melting point of ITZ. Typically, in HME, the temperature of the molten zone is set 15–60 °C higher than the glass transition temperature or melting point of the carrier. However, to ensure the physical stability of the drug, it can be set at a lower temperature (Appendix A). The feasibility of preparation at these lower temperatures is due to the mechanical energy provided by the design of the screw and the shear pressure [34,35]. The extrudates were subjected to a cooling period of 30 min at 25 °C, followed by a primary milling process for 5 min using a mortar and pestle. Subsequently, jet milling was performed at a grinding pressure of 0.60 MPa and a supply pressure of 0.65 MPa. After extrusion, the jet-milled ITZ microparticles were collected in glass bottles and stored in a desiccator with silica beads until further use. These preparations are referred to as HME formulations.

### 2.3. Physicochemical Characterization

#### 2.3.1. Particle Size Distribution (PSD) by Laser Diffraction

The particle size distributions of the JMs and HMEs were measured using a laser diffraction particle size analysis using the dry dispersion method (Mastersizer 3000 AERO S, Malvern Panalytical Ltd., Malvern, UK). Each measurement was conducted in triplicate, and the mean and standard deviation (SD) were calculated. The samples were pretreated by sieving through a 600 µm mesh sieve to break up excessive aggregation just before measurement. A dry-powder feeder was used at a pressure of 4 bar and a feed rate of 20%. These conditions allowed the measurement of the PSD of the almost completely deagglomerated powder owing to the highly intensive dispersion conditions generated in the dispersion unit. Dv (10), Dv (50), and Dv (90), which are diameters at 10%, 50%, and 90% cumulative volume, were calculated from the size distribution data. Dv (50) represented the median particle size of the volume distribution. Span, a measure of the polydispersity of the PSD, was defined using the following equation:$$\mathrm{Span}=\frac{\mathrm{Dv}\;(90)-\;\mathrm{Dv}\left(10\right)}{\mathrm{Dv}\;(50)}$$

#### 2.3.2. Scanning Electron Microscopy (SEM)

The particle and surface morphologies of the microparticles were investigated using an ULTRA PLUS SEM (Zeiss Group, Jena, Germany). Samples were attached to carbon tape on an aluminum plate and coated with a gold film (600 Å) using sputter deposition techniques. Observations were conducted at a voltage of 3 kV and magnifications of 250× and 1000×. Visualization was performed using a SE-2 detector.

#### 2.3.3. Thermal Analysis Differential Scanning Calorimeter (DSC)

The thermal response of raw materials and each DPI formulation was analyzed using a DSC Q2000 (TA Instruments^®^, New castle, DE, USA) thermal analyzer system. Samples (5 ± 2 mg) were accurately weighed, loaded in an aluminum pan, and analyzed at a heating rate of 10 °C/min over a temperature range of 0 °C to 300 °C. The data were analyzed with TA Universal Analysis^®^ software version 5.2.6 (TA Instruments^®^, Delaware, DE, USA).

#### 2.3.4. Powder X-ray Diffraction Analysis (PXRD)

The PXRD patterns of the raw materials and each DPI formulation were analyzed using SmartLab with an SC-70 detector (Rigaku Corporation, Osaka, Japan). Measurements were conducted with Cu Kα radiation (λ = 1.540 Å) at a voltage of 45 kV and a current of 200 mA. The 2θ scan range was from 5° to 40° with a step size of 0.02° and a scan speed of 5°/min.

### 2.4. In Vitro Aerodynamic Performance Study

In accordance with the USP Chapter <601> specifications for aerosols, the aerodynamic performance of the JM and HME formulations was evaluated using a next-generation impactor (NGI, Copley Scientific Limited., Nottingham, UK) and RS01 DPI device. To prevent particle bounce and re-entrainment, the collection plates of the NGI stages were pre-coated with 3% silicone oil in hexane. Each sample containing 10 mg of ITZ was loaded into hydroxypropyl methylcellulose hard capsules (size 3). A capsule was inserted into the RS-01, and the device was inserted into the mouthpiece of the induction port. Air was inhaled at a controlled flow rate of 60 L/min for 4 s. For an NGI flow rate of 60 L/min, the aerodynamic cutoff diameters of each stage were determined as 8.06 µm, 4.46 µm, 2.82 µm, 1.66 µm, 0.94 µm, 0.55 µm, 0.34 µm, and 0.14 µm for stages 1–7 and Micro Orifice Collector (MOC). The quantity of the sample remaining in the capsule and deposited on each collection plate at each stage was quantified using HPLC. The aerodynamic performance tests were performed in triplicate. The emitted dose (ED), fine particle fraction (FPF), and extra fine particle fraction (eFPF) were calculated using the following equations:

Example of Equation:Emitted Dose [ED, %] = [(Initial Mass in Capsule − Final Mass Remaining in Capsule)/(Initial Mass in Capsule)](1)
Fine Particle Fraction [FPF, %] = [Mass of Particles in Stages 2 − MOC]/[Mass of Particles in All Stages] (2)
Extra Fine Particle Fraction [eFPF, %] = [Mass of Particles in Stages 4 − MOC]/[Mass of Particles in All Stages] (3)

The mass median aerodynamic diameter (MMAD) and geometric standard deviation (GSD) were calculated using the guidelines provided in USP Chapter 601. The MMAD was determined from a plot of a mass fraction smaller than the aerodynamic diameter, specified as D50%, on a logarithmic probability scale. The GSD was calculated using the following equation:GSD = √[D84.13%/D15.87%](4)

### 2.5. In Vitro Dissolution Behavior and Solubility

#### 2.5.1. Solubility

The volume of lung fluid is extremely limited and presents a non-sink condition [36]. To assess the potential of providing similar to lung fluid, the saturation solubility kinetics of the formulation were determined. A developed supersaturated solution in a medium dissolution medium, referred to as “Simulated Lung Fluid” (SLF), was described, featuring ionic composition and pH close to that of lung fluid. However, one study observed a rapid increase in the pH from 7.2 to 8 within 2 h, which was attributed to the poor buffering capacity of the SLF. Because of the significant impact of the pH on the solubility of the ITZ, this dissolution medium was deemed unsuitable [37]. Instead, the physiological buffer described in the European Pharmacopoeia 7.2 (adjusted to pH 7.2) was used. This “physiological” solution comprised 0.8% sodium chloride, 0.02% potassium chloride, 0.01% calcium chloride, 0.01% magnesium chloride, 0.318% disodium hydrogen phosphate, and 0.02% potassium dihydrogen phosphate (*w*/*v*). Given the highly insoluble nature of ITZ, 0.3% (*w*/*v*) Sodium lauryl sulfate (SLS) was added to this solution to enhance the analytical concentration. In a conical tube containing 10 mL of physiological solution with 0.3% SLS, excess amounts of raw ITZ, JMs, and HME160s samples were added to induce a supersaturated state. The solubility tests were conducted using a Shaking Heating Bath (BW-05G, Jeio tech, Daejeon, Republic of Korea), maintaining the medium temperature at 37.0 ± 0.5 °C and stirred at 50 rpm. Subsequently, 1 mL samples were collected after 360, 720, and 1440 min and directly filtered through a 0.45 µm PVDF filter to avoid quantification of undissolved particles. Samples were reconstituted in ACN for HPLC. The solubility tests were performed in triplicate.

#### 2.5.2. In Vitro Release Study

The release rate of the ITZ from the samples was investigated using a Franz diffusion cell. This release study was conducted to compare the dissolution rates based on the proportion of LEU and incorporation of the HME process. To evaluate the drug release from particles reaching the cardiopulmonary system, DPI particles with an aerodynamic diameter between 2.82–4.46 μm were collected from the stage 3 plate of a NGI and then introduced into the dissolution media. The dissolution media employed 70% (*v*/*v*) EtOH to meet sink conditions, and a volume of 15 mL was used, set based on the estimated volume of lung fluid. This media was maintained at a temperature of 37 °C and stirred at a speed of 400 rpm. Each sample containing 500 μg of ITZ was distributed on a 0.45 μm regenerated cellulose membrane filter. Sampling was carried out at intervals of 15, 30, 60, 120, 240, and 360 min, with each sampling volume being 400 μL. To maintain a constant volume of the elution medium, the solvent removed during sampling was replaced with fresh preheated elution media. The ITZ content was determined by HPLC, and each test was performed in triplicate.

#### 2.5.3. Dynamic Vapor Sorption (DVS)

The vapor adsorption of raw ITZ, JMs, and HME160s was performed using a Dynamic Vapor Sorption (DVS) intrinsic automatic gravimetric sorption analyzer (Surface Measurement Systems, Ltd., London, UK). Samples weighing 15 ± 2 mg were placed in a net basket and placed within the system. The samples were equilibrated at an analysis temperature of 25 °C and a Relative Humidity (RH) of 0%. Subsequently, they were exposed to an RH profile increasing in increments of 10% within the range of 0% to 95%. Equilibrium of the sample moisture mass was achieved at each stage before changing the RH, defined as dm/dt of 0.0001% per minute. The amount of absorbed water was expressed as a percentage of the reference mass.

#### 2.5.4. Contact Angle

The contact angles were measured to study the wettability in water. The contact angles of raw ITZ, JMs, and HMEs were measured using a contact angle analyzer (Phoenix 300-Touch, SEO Co., Suwon, Republic of Korea). Approximately 100 mg of DPI powder was compressed into a slice for one minute (35% RH and 25 °C) using a press. Using a vertical electronic syringe, a drop of DW (5 μL) was injected onto an 8.0 mm^2^ surface of the sample piece from a height of 0.45 mm above the base. The contact angle was measured 5 s after dropping. The contact angle, which is defined as the tangent at the base of the droplet, was measured in degrees. Each sample was measured thrice to obtain an average value.

### 2.6. Raman Microscopy

A RAMANwalk (Nanophoton Corporation, Osaka, Japan), a random scanning Raman microscope equipped with a 532 nm diode laser, was used to acquire individual Raman reference spectra from single components and collect Raman image data from powder mixtures. The powder was spread evenly on a glass microscope slide to create a flat and smooth surface. Spectra were collected using X-Y random mapping at 50× magnification. The measurement range was 130 µm horizontally and 100 µm vertically. The ND filter was set at an intensity of 3.98% (120/255). Images were obtained with a 1 s exposure and three cumulative scans. Raman images were generated from the spectra collected over approximately 44 h using the direct classical least-squares method. In the Raman image, ITZ, MAN, and LEU are indicated in red, green, and blue, respectively. The area divisions of the ITZ and LEU were obtained using scatter analysis. The binarized image was divided into squares of 5 μm width, and the number of white pixels in each area was counted. The value obtained by dividing the standard deviation of the number of white pixels in each section by the average score was calculated to evaluate the extent to which the ingredients are aggregated quantitatively. The more aggregated the components are, the higher the score.

### 2.7. HPLC Assay

The HPLC method for the quantitative analysis of the ITZ was conducted using a Thermo Ultimate 3000 HPLC system (Thermo Scientific, Waltham, MA, USA). The column was Inertsil ODS-2 4.6 × 250 mm, 5 µm HPLC analytical column (GL Sciences, Tokyo, Japan). The mobile phase was composed of a mixture of buffer containing 27.2 mg of tetrabutylammonium hydrogen sulfate dissolved in 1000 mL of water buffer and ACN in a ratio of 30:70 (*v*/*v*) and was filtered through a 0.45 µm membrane filter and then degassed before use. The mobile phase was pumped through the column at a flow rate of 2.0 mL/min. The column temperature was set to 30 °C, and the detection wavelength was 225 nm. The injected volume of each sample was 20 µL. The HPLC retention time was 10 min, and ITZ was detected at 6.0 min. The standard curve was taken at eight points, 100, 50, 25, 12.5, 6.25, 3.125, 1.5625, and 0.78125 µg/mL, based on dilutions of the standard solution of ITZ at 100 µg/mL. The r^2^ value of the standard curve is 0.99998. The HPLC method was validated for the calculations.

### 2.8. Statistical Analysis

Statistical significance was assessed using one-way analysis of variance (ANOVA) with Tukey’s test in Prism version 8 (GraphPad Software, San diego, CA, USA). Statistical significance was set at *p* < 0.05. All data are expressed as the mean + standard deviation (SD) where n  ≥  3.

## 3. Results and Discussion

### 3.1. Physicochemical Characterization

#### 3.1.1. Particle Size Distribution (PSD) Using Laser Diffraction

The particle size distributions of the formulations are summarized in Table 2. All the formulations exhibited unimodal and narrow particle size distributions [38]. There were no significant differences in particle size distribution among the formulations. Specifically, Dv (10) was approximately 1 µm, Dv (50) ranged from 3.21 to 3.99 µm, and Dv (90) ranged from 6.24 to 8.45 µm across all formulations. This indicates that the introduction of HME or LEU did not result in differences in the physical particle size distribution. The key point here is that while the physical particle size distribution did not show significant differences, as mentioned later, the in vitro aerodynamic performance study demonstrated significant differences.

Nevertheless, the trends were observed to depend on the LEU ratio. Compared to the L0 formulation without LEU, both L0.1 and L1 formulations showed a slight increase in Dv (10) as the ratio of LEU increased. However, in the L10 formulations, both JM and HME showed a slight decrease in Dv (10). Similarly, Dv (50) and Dv (90) also showed a trend of increasing particle size with an increasing ratio of LEU, with a consistent trend of particle size reduction observed at a 10% LEU ratio. These trends suggest that the addition of LEU did not significantly alter particle size up to 1%, showing only a slight increasing trend. However, at a 10% LEU ratio, some particle aggregation seemed to be reduced, leading to a trend of decreasing particle size. These trends indicate the potential surface modification properties of LEU. These surface modification properties were further confirmed through contact angle studies, which provided more direct evidence of their effects on L10 [39,40,41].

#### 3.1.2. Morphology

The morphologies of the samples are shown (Figure 1). The ITZ exhibited needle-shaped, thin, and elongated particles, whereas MAN showed particles with angular edges, and LEU displayed rounded and broad plate-like structures [42]. In JMs, the physical forms of the MAN and LEU are maintained, allowing particle differentiation through visual inspection [43]. In JM-L0 and JM-L0.1, particle agglomeration was observed. These agglomerates can impede dispersion within the desired inhalable size range [44]. In HMEs, owing to the melt-extrusion process, the original particle shapes of the individual components were indiscernible. The particles formed new and distorted shapes, and smaller particles were observed compared to those in the JMs. The particles in each HME formulation were morphologically similar. HME-L0, HME-L0.1, HME-L1, and HME-L10 all exhibited comparable levels of aggregation and surface adhesion between particles. Consistent with the PSD results, all formulations displayed similar particle size ranges. Furthermore, similar to the Span values, there was a tendency for the particle size distribution to broaden with increasing LEU rato in the HME formulations.

#### 3.1.3. Crystallinity

Crystal-state information was obtained using DSC and PXRD. In the DSC thermal analysis diagram (Figure 2), the primitive ITZ exhibits a distinct endothermic peak at 168 °C, whereas MAN exhibits a broad endothermic peak at 167 °C [45]. The thermal behavior of the JMs and HMEs, characterized by an endothermic peak similar to that of the primitive ITZ, did not show significant differences in thermal properties. Therefore, the ITZ in JMs and HMEs were crystalline [23]. The amorphous form is inherently physically unstable, and exposure to ambient moisture can decrease the powder dispersibility and reduce the aerodynamic performance, potentially leading to recrystallization tendency [46]. The crystalline form of the ITZ is physically more stable and can enhance the powder dispersibility and aerodynamic performance [45]. These endothermic peaks represent the melting points and are consistent with the melting points of ITZ and MAN as reported in previous studies [23]. For LEU, no peaks were observed up to 200 °C. According to the literature, the melting point of LEU is approximately 286 °C [47].

As depicted in Figure 3, the PXRD patterns of PMs, JMs, and HMEs exhibited characteristic peaks of ITZ at 9.00, 11.50, 18.00, and 19.80°, and specific peaks of β-MAN at 10.60, 14.80, 23.50, 29.80, 39.00° [48]. These results indicate the crystalline nature of both ITZ and MAN. However, the crystallinity of the HMEs decreased, as indicated by the reduced intensity of the characteristic ITZ peaks. In particular, in HME-L0 and HME-L10, the intensity of the ITZ peaks was lower compared to HME-L0.1 and HME-L1 [48]. The β form was more thermodynamically stable than the α form. The α form is characterized by its tendency to generate cohesive and adhesive powder [49]. Thus, β-MAN could be more advantageous in terms of stability and aerosolization of DPI microparticles. Specific peaks of LEU were observed at 12.40, 24.60, 30.80, and 37.20° in the patterns of PM-L1 and PM-L10. However, in PM-L0.1, no specific peak of LEU was observed, indicating that the low concentration of LEU was insufficient to produce a detectable peak. In the L1 and L10 patterns of JMs and HMEs, the absence of specific LEU peaks suggests a reduction in crystallite size due to the milling process [50].

### 3.2. In Vitro Aerodynamic Performance Study

Aerodynamic performance is influenced by various physicochemical properties such as particle size, density, surface characteristics, and crystallinity. Therefore, in vitro aerodynamic performance studies can be interpreted in association with physicochemical characterization studies [51,52,53]. 

The aerodynamic performance characteristics were evaluated based on the concentration of LEU and the implementation of the HME process. Figure 4 shows the proportion of ITZ deposited at each stage during the assessment of in vitro aerodynamic performance using NGI. Both the JMs and HMEs demonstrated the least residual ITZ in the device and IP (throat) at L10, with increased deposition of the ITZ in the latter stages, characterized by smaller cutoff diameters. Furthermore, the HMEs showed greater ITZ deposition from stage 4 compared to the JMs. The deposition from stage 2 to the MOC is considered FPF, and that from stage 4 to the MOC is considered eFPF [54]. ED denotes the emitted dose relative to the initial amount contained in the capsule. FPF refers to the proportion of particles with an aerodynamic diameter of less than 8.06 µm, which theoretically represents the size range capable of reaching the lungs. eFPF represents the proportion of particles with a diameter smaller than 2.82 µm. Recent studies have indicated that eFPF provides a higher predictive value for clinical lung deposition compared to FPF [55,56]. In JMs, an increase in LEU concentration led to an improvement in ED and FPF values, although the difference was not statistically significant. No trend was observed in the eFPF of the JMs. However, in HMEs, an increase in LEU concentration resulted in a gradual increase in both ED and FPF, as well as in eFPF (Table 3). These results were supported by contact angle studies, which attributed them to the powder dispersion effect and lubricating ability of LEU [31]. Comparatively, the ED (%) was significantly higher in L0 and L0.1 of HMEs than in JMs, at 84.95 ± 2.19% and 91.45 ± 1.66%, respectively. Also, in eFPF (%), L1 and L10 of HMEs exhibited significantly higher values compared to JMs, at 21.34 ± 2.94% and 33.68 ± 1.31% respectively. This suggests that the ability to disperse small amounts of LEU through HME contributes to these results. Among all formulations, HME-L10 had the highest ED, FPF, and eFPF values, with an MMAD of 3.46 ± 0.06 μm, substantially smaller than other formulations. This indicated the most successful delivery of ITZ to the deep regions of the lungs [54,57]. Furthermore, formulations containing a slight amount of LEU, such as L0.1 and L1, tended to exhibit decreased eFPF values compared to the LEU-deficient formulation L0. Conversely, formulations with a 10% LEU addition, like L10, demonstrated a notable tendency toward increased eFPF values. This trend aligns with the PSD patterns. Considering that the flow-dependent mechanism is influenced by particle size, these variations among the formulations can be attributed to the aerodynamic diameter [58]. These results demonstrate the contribution of the LEU and HME processes in enhancing aerodynamic performance. HME-L10, which exhibits sufficient miscibility through HME and thereby provides significant hydrophobic action on the surface due to LEU, shows noticeably higher eFPF values compared to other formulations. Particularly, the deposition differences are pronounced in stages 5 and 6. This indicates an increase in particles with an aerodynamic diameter ranging from 0.55 to 1.66 µm, suggesting a higher potential for drug delivery deep into the alveolar regions beyond the airway distribution. Additionally, these ultrafine particles are expected to result in faster drug absorption in the alveoli [59]. In conclusion, the efficacy of ITZ microparticles for efficient inhalation and pulmonary deposition has been validated.

### 3.3. In Vitro Dissolution Behavior and Solubility

#### 3.3.1. Solubility

The solubility of ITZ in raw form and each DPI formulation was measured at pH 7.2 using 0.3% (*w*/*v*) SLS, as depicted in Figure 5. Compared with raw ITZ, both JMs and HMEs exhibited enhanced solubility. The difference in solubility between raw ITZ, JMs, and HMEs was statistically significant. This increase can be attributed to the reduction in the particle size achieved by jet milling [38]. As the particle size decreases, the surface area of the particles increases, leading to enhanced solubility [60]. HMEs showed approximately 1.5 times higher solubility than JMs, and this difference was statistically significant. The enhanced solubility of HMEs appears to be due to two factors that result from the effects of the HME. Solubilization by HME leads to a decrease in crystallinity, which in turn contributes to increased solubility [61,62]. The highest solubility was observed for HME-L10 and HME-L0, both of which exhibited the lowest intensity of ITZ peaks in the PXRD analysis. The HME process distributes the ITZ within the hydrophilic MAN, creating a solid crystalline suspension (crystalline drug suspended in a crystalline carrier matrix). This solid crystalline suspension was designed to improve the dissolution of poorly soluble drugs [63,64]. Moreover, the elevated processing temperatures and rapid screw speeds characteristic of the HME process are pivotal for augmenting solubility [63,65].

#### 3.3.2. In Vitro Release Study

The volume of alveolar fluid is minimal, and there is no standardized method for measuring the leaching behavior of inhaled dry powder [66]. The Franz diffusion cell provides an air-liquid interface similar to that of the lungs and is suitable for depicting a limited amount of alveolar fluid [67]. 

Considering the concentration of LEU and the introduction of the HME process, the drug release rate over 6 h under sink conditions was compared using a Franz diffusion cell (Figure 6). Neither JMs nor HMEs showed a trend in the release rate depending on the concentration of LEU. The initial release rates were observed in relation to the solubility results, which depended on the presence of HME. As indicated by the solubility results, HMEs had higher solubility than JMs. Following this trend, HMEs exhibited a faster release rate than JMs.

Figure 6A illustrates the drug release profile of the LEU-deficient formulation L0. The average cumulative drug release at the final sampling point at 6 h increased by approximately 15%, with JM-L0 at 37.2% and HME-L0 at 52.3%. Figure 6B represents the drug release profile of the formulation L0.1 with 0.1% LEU addition. Surprisingly, it exhibited an average cumulative drug release at 6 h, with the same increase rate as L0, approximately 15%. Specifically, JM-L0.1 recorded a drug release of 25.4%, and HME-L0.1 recorded 40.2%. Figure 6C illustrates the drug release profile of the formulation L1 with a 1% LEU addition. The average cumulative drug release at 6 h increased by approximately 32%, with JM-L1 at 35.5% and HME-L1 at 67.8%, demonstrating the highest increase in drug release rate among HME formulations. Figure 6D displays the drug release profile of the formulation L10 with a 10% LEU addition. The average cumulative drug release at 6 h increased by approximately 21%, with JM-L10 at 39.4% and HME-L10 at 60.2%, indicating the second-largest increase in drug release rate among HME formulations.

Among the HME samples, HME-L1 exhibited the fastest and highest drug release rates, followed by HME-L10. Although HME-L10 showed the highest solubility, its drug release rate was slightly lower than that of HME-L1. This suggests the possibility of surface modification by LEU, a phenomenon that was directly confirmed through subsequent wettability studies. Briefly, in the L10 formulation, the properties of LEU were effectively exhibited, leading to a reduction in surface energy and wettability, which are believed to have partially affected the dissolution rate.

In summary, the solubility enhancement effect of HME further promotes drug dissolution and dissolution rate [65]. Despite the hydrophobic surface properties imparted by LEU, the high miscibility of HME ensured uniform distribution of LEU, resulting in a faster dissolution rate for HME-L10 compared to JM-L10. Considering the rapid onset effect of inhalers, the initial dissolution times of 15 min, 30 min, and 60 min showed more pronounced differences between JM-L10 and HME-L10 than formulations with other LEU ratios. Additionally, the 15-min dissolution rate was highest in HME-L10 compared to all other formulations. This suggests that the superior miscibility of HME allows for effective hydrophobic action of LEU on the surface during storage and inhalation, while also ensuring uniform distribution of hydrophilic MAN, enabling rapid dissolution at the deposition site. These findings imply a rapid onset of drug action at the disease site.

#### 3.3.3. Dynamic Vapor Sorption (DVS)

Moisture absorption influences drug dispersion and deagglomeration, thus affecting the performance of respiratory drug delivery [68]. The DPI’s delivery performance appears to be influenced by storage humidity conditions [69]. 

In Figure 7A, comparisons were made among the moisture adsorption trends of raw ITZ, JM-L0, and JM-L10. The insoluble raw ITZ exhibited the lowest moisture adsorption and desorption. At 95% RH, the weight change of raw ITZ was only 0.1687%, indicating that hydrophobic ITZ absorbed minimal water from humid air even under high RH conditions. The weight changes of JM-L0 and JM-L10 at 95% RH were 0.3452% and 0.3216%, respectively. Both JM-L0 and JM-L10 presented higher moisture adsorption rates than raw ITZ. This appeared to be due to the influence of hydrophilic excipients and the increased surface area resulting from particle size reduction achieved via jet milling [70]. Figure 7B compares the moisture adsorption trends of HME-L0 and HME-L10 with those of raw ITZ. HME-L0 and HME-L10 exhibited maximum weight changes of 0.5538% and 0.4747%, respectively. This represents a weight change significantly higher than that of raw ITZ, with increases of 3.3-fold and 2.8-fold, respectively. They demonstrated a higher weight change tendency than the JM formulations with the same LEU ratio. 

Comparison between JM-L0 and HME-L0 revealed the effects of HME. The HME exhibited faster moisture adsorption rates and a larger hysteresis gap. This suggests that while JM, which maintains more crystallinity, primarily adsorbs moisture only at the surface, HME adsorbs moisture not only at the surface but also within the powder, resulting in the observed hysteresis gap [71,72].

HME-L0 and HME-L10 demonstrate the effects of LEU well. The addition of LEU to HME-L10 resulted in slower moisture adsorption rates and a smaller hysteresis gap than those of Leu-free HME-L0. It is inferred that with sufficient and homogeneous incorporation of LEU, the protective effect of LEU on the surface against moisture is evident. Consequently, it is presumed that preventing the penetration of moisture into the powder interior reduces the hysteresis gap [73,74].

Overall, no significant difference in absorption rate was observed in JM formulations based on the presence or absence of LEU, whereas HME-L10 exhibited lower adsorption rates compared to HME-L0 when 10% LEU was added. Considering typical storage conditions, the moisture absorption rates of HME-L0 and HME-L10 at 60% RH are 0.31% and 0.23%, respectively. HME-L0 exhibits a relatively wider hysteresis range compared to HME-L10 and other formulations. During the SD process, LEU migrates to the surface as drying occurs. However, in the HME process, the material is dried at high temperatures in the form of filaments. This results in sufficient LEU being positioned on the surface due to the high miscibility of HME, rather than surface migration. The strongly adhered hydrophobic properties of LEU are believed to provide enhanced protection against moisture. Consequently, a small hysteresis gap in the adsorption–desorption isotherms was observed for all the samples. All DPI formulations demonstrated low hygroscopicity and excellent physicochemical stability at all tested RH levels.

#### 3.3.4. Contact Angle

As shown in Figure 8, the water contact angles of raw ITZ, JM, and HME were measured to evaluate the wettability of the formulation surface. An increase in the contact angle signifies a decrease in wettability, indicating a decrease in the surface energy. Contact angle studies allow for a more concentrated analysis of surface characteristics than DVS analysis, providing direct insights into surface energy [75,76,77,78].

The measurements were performed on a smooth surface, as shown in Figure 8. The contact angle of the raw ITZ was high at 71.93 ± 4.54°, which is expected to be due to its strong hydrophobic nature. In most JM and HME formulations, a lower contact angle was observed compared to that of raw ITZ, indicating the presence of the hydrophilic excipient MAN. No clear trend in contact angle was observed for JM using different concentrations of LEU. In contrast, in HME, an increase in LEU concentration corresponded to an increase in the contact angle, which for HME-L10 reached levels similar to those of raw ITZ. Additionally, it was observed that HMEs with the same ratio of LEU tended to have higher contact angles with JMs. This indicates that even though the solubility of HMEs was approximately 1.5 times higher than that of JMs, sufficient surface modification was achieved through LEU. Through homogeneous and rapid mixing, formulations containing LEU in HME are expected to have reduced surface wettability compared to JM. An increase in contact angle is associated with a decrease in surface energy. This can be expected to improve moisture protection and dispersibility during storage and increase aerodynamic performance [78,79,80].

### 3.4. Raman Imaging and Dispersibility

Dispersiveness analysis evaluates the heterogeneity, cohesiveness, and deviation of the distribution of each component in Raman imaging [81,82]. The dispersion of the ITZ and LEU in various samples was observed using Raman imaging (Figure 9A,B). The aggregation of ITZ and LEU in the JMs was observed using Raman binary imaging. Additionally, in the HMEs, a more uniform dispersion of ITZ and LEU with smaller particle sizes was observed compared to the JMs. This difference was particularly pronounced in HME-L10. Figure 9C,D shows the dispersion analysis methods quantitatively. The dispersibility of the ITZ showed statistically significant lower area fractionation in HME-L0.1 and HME-L10. Conversely, the dispersibility of LEU showed a statistically significant lower area fractionation in HME-L10, indicating superior dispersion results in HME-L10. These results confirmed the enhanced dispersion characteristics of HME owing to the more uniform mixing of melt-extruded particles, ultimately improving the inhalation efficiency of the formulation [23,83,84]. In particular, HME-L10 demonstrated sufficient surface modification and effective deagglomeration owing to the adequate introduction of LEU and the homogeneous effect of HME. This was confirmed in an aerodynamic performance study that described an increase in eFPF, a metric indicating delivery to deeper lung regions. Nevertheless, there are limitations in discussing uniformity based solely on imaging studies. Imaging studies have been conducted on a very narrow range of formulations, making it challenging to generalize the uniformity across batches.

## 4. Conclusions

Inhalable particles of the ITZ containing LEU were prepared with jet milling or a combination of HME and jet milling. Changes in the physical and chemical properties of particles produced using various preparation methods were observed. As a result, a partial decrease in crystallinity was a key characteristic of physicochemical changes. 

Furthermore, owing to the solubilizing effect of HME, the solubility increased by approximately 1.5 times compared with JM, with a corresponding increase in the dissolution rate. Additionally, DVS confirmed that HME particles with reduced crystallinity exhibited higher hygroscopicity than JM particles. Nevertheless, an increase in the ratio of LEU in the HME formulations resulted in a decrease in surface wettability. Although the solubility increased, a decrease in wettability indicated surface modification. This, in turn, could have a positive impact on the dispersibility and aerodynamic performance owing to a decrease in the surface energy of the particles. Through Raman imaging, we quantitatively and qualitatively confirmed the improvement in the dispersibility of the ITZ and LEU as the LEU increased. Through contact angle and Raman imaging studies, it was confirmed that the rapid and homogeneous mixing of HME maximizes the surface modification effect and dispersibility enhancement of LEU.

Therefore, the variations mentioned above in the characteristics according to the ratio of LEU provided insights into the aerodynamic behavior and particle size distribution, demonstrating an enhancement in the aerodynamic performance. The addition of HME maximized the effects of LEU and improved the dispersibility using homogeneous mixing. This study verified the efficient surface modification achieved by combining LEU and HME, indicating the potential for effective pulmonary delivery of poorly soluble drugs such as ITZ.

## Figures and Tables

**Figure 1 pharmaceutics-16-00784-f001:**
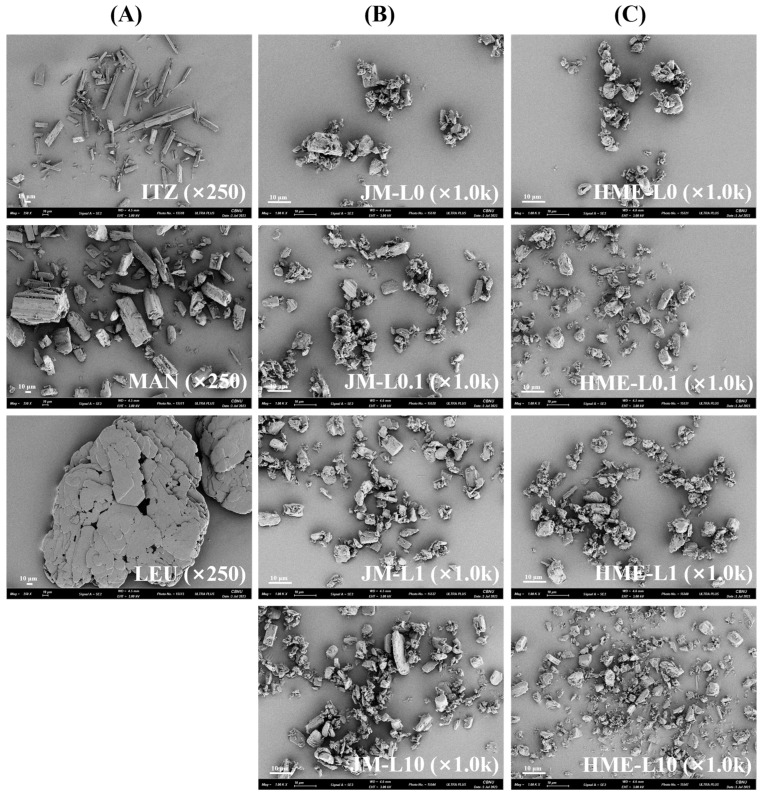
SEM micrographs of (**A**) raw materials, (**B**) JMs, and (**C**) HMEs. Magnification for samples was 250× and 1000×, respectively.

**Figure 2 pharmaceutics-16-00784-f002:**
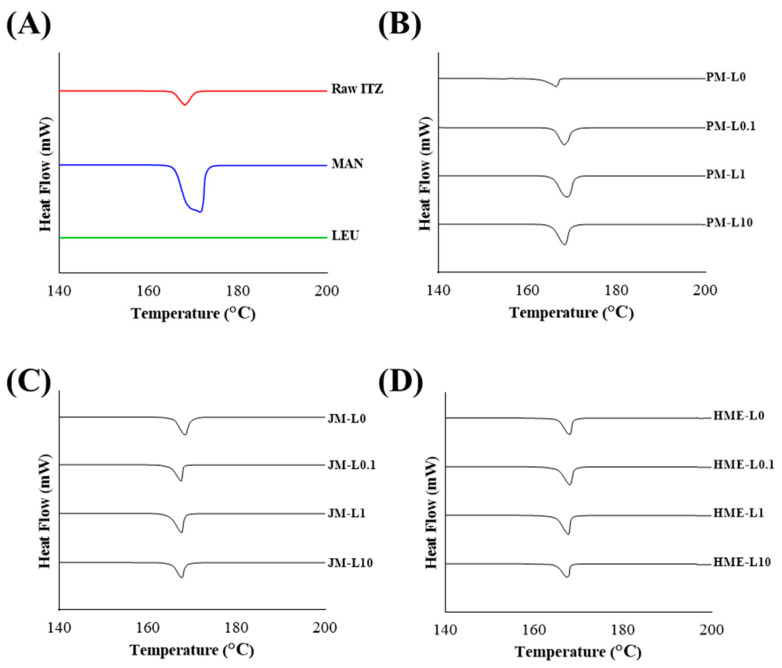
DSC thermogram of (**A**) raw materials, (**B**) PMs, (**C**) JMs, (**D**) HMEs.

**Figure 3 pharmaceutics-16-00784-f003:**
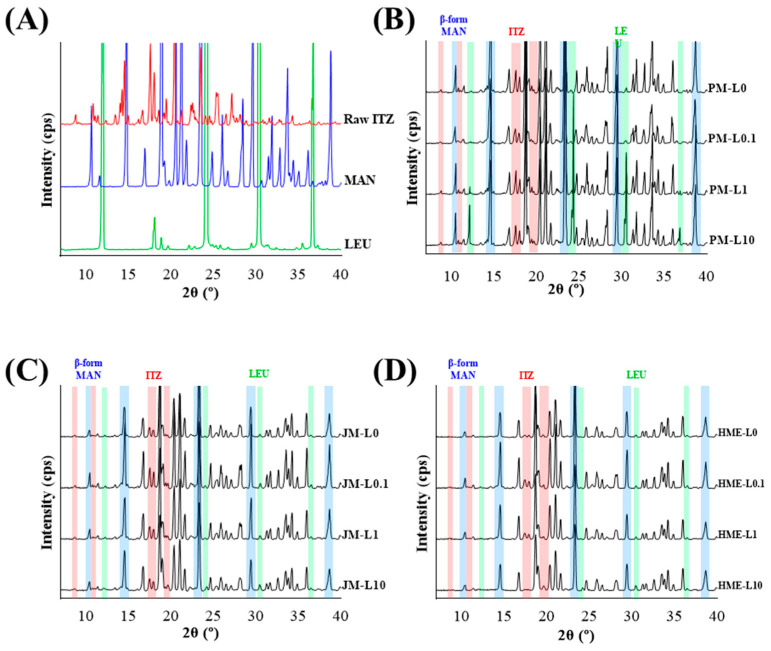
XRD pattern of (**A**) raw materials, (**B**) PMs, (**C**) JMs, and (**D**) HMEs. Colors represent the following: ITZ is red, MAN is blue, and LEU is green.

**Figure 4 pharmaceutics-16-00784-f004:**
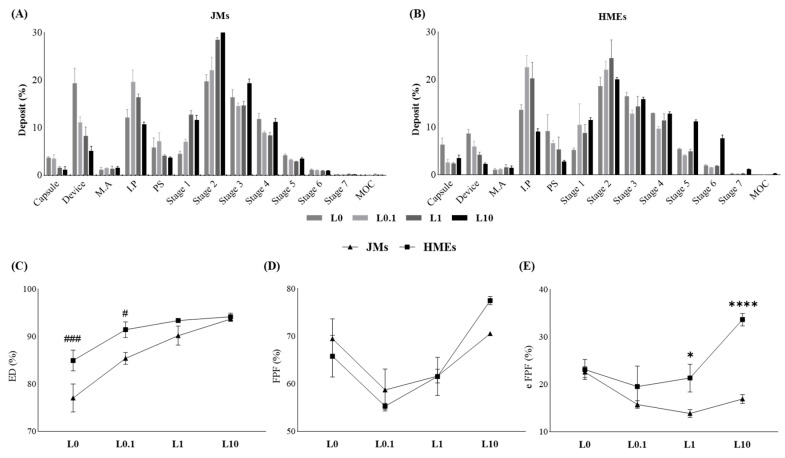
Percentage deposition in each stage of NGI: (**A**) JMs and (**B**) HMEs. Aerodynamic performance according to LEU ratio in NGI: (**C**) ED, (**D**) FPF, and (**E**) eFPF (mean ± SD, n = 3). ^###^ ANOVA, *p*-value < 0.001 compared with JM-L0; ^#^ ANOVA, *p*-value < 0.05, compared with JM-L0.1; * ANOVA, *p*-value < 0.05 compared with JM-L1; **** ANOVA, *p*-value < 0.0001 compared with JM-L10.

**Figure 5 pharmaceutics-16-00784-f005:**
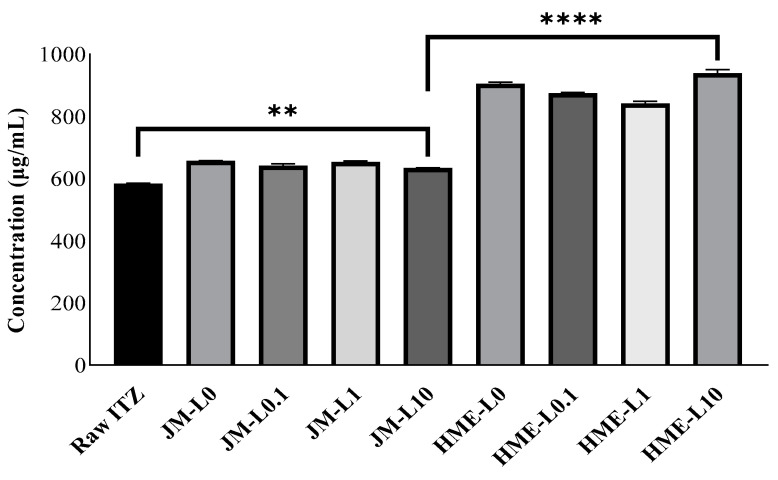
Solubility of raw ITZ, JMs, HMEs (mean ± SD, n = 3). **** ANOVA: HME-L10 vs. JM-L10, *p*-value < 0.0001; ** ANOVA: JM-L10 vs. raw ITZ, *p*-value 0. 0035.

**Figure 6 pharmaceutics-16-00784-f006:**
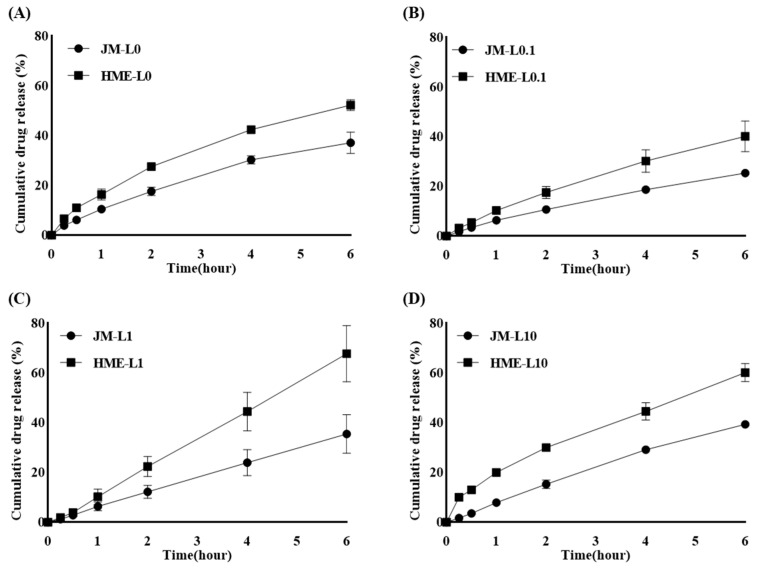
The in vitro drug release profiles of JMs and HMEs according to LEU ratios in Franz diffusion cells: (**A**) L0, (**B**) L0.1, (**C**) L1, and (**D**) L10 (mean ± SD, n = 3).

**Figure 7 pharmaceutics-16-00784-f007:**
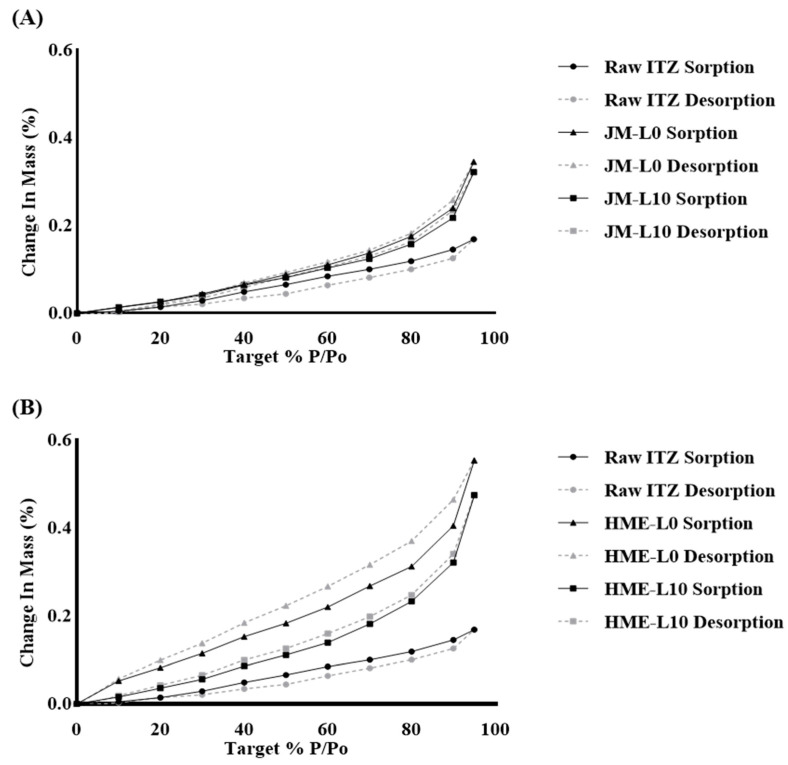
Water sorption/desorption isotherms for ITZ and formulations at 25 °C: (**A**) JM-L0, JM-L10, and (**B**) HME-L0, HME-L10.

**Figure 8 pharmaceutics-16-00784-f008:**
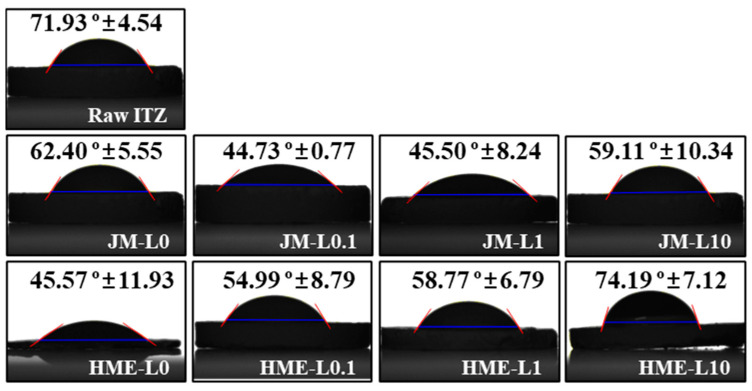
Comparison of wettability through contact angle measurement (mean ± SD, n = 3).

**Figure 9 pharmaceutics-16-00784-f009:**
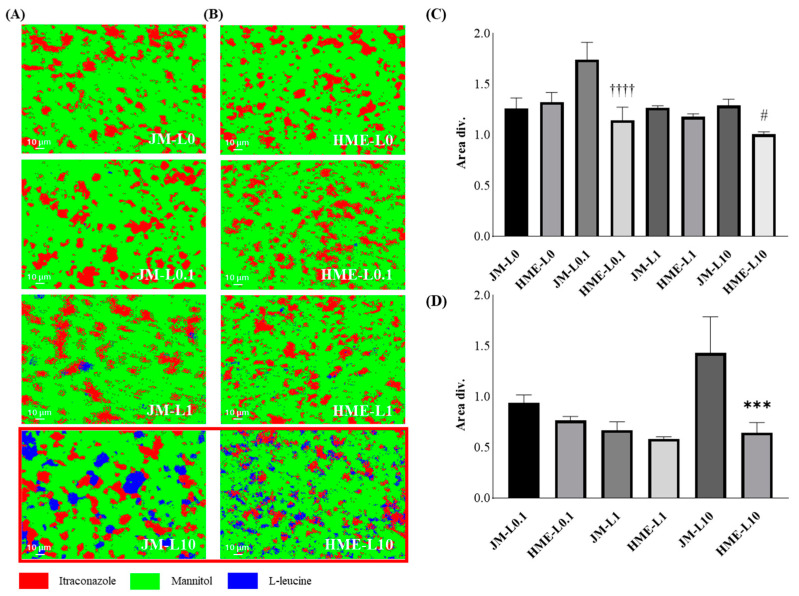
Binary images of Raman microscopy: (**A**) JMs and (**B**) HMEs. Dispersiveness analysis of Raman microscopy: (**C**) ITZ dispersibility of JMs and HMEs, and (**D**) LEU dispersibility of JMs and HMEs (mean ± SD, n = 3). ^††††^ ANOVA, *p*-value < 0.0001 compared with JM-L0.1; ^#^ ANOVA, *p*-value < 0.05 compared with JM-L10; *** ANOVA, *p*-value < 0.05 compared with JM-L10.

**Table 1 pharmaceutics-16-00784-t001:** Formulation of ITZ microparticles.

Code	Process	Formulation Ratio (%)		
		Itraconazole (ITZ)	Mannitol (MAN)	L-Leucine (LEU)
JM-L0	Co-jet mill	20.00	80.00	-
JM-L0.1		19.98	79.92	0.10
JM-L1		19.80	79.20	1.00
JM-L10		18.00	72.00	10.00
HME-L0	Jet mill after HME	20.00	80.00	-
HME-L0.1		19.98	79.92	0.10
HME-L1		19.80	79.20	1.00
HME-L10		18.00	72.00	10.00

**Table 2 pharmaceutics-16-00784-t002:** Particle size distribution of JMs and HMEs.

Formulation		Dv (10) (μm)	Dv (50) (μm)	Dv (90) (μm)	Span
Co-jet mill	JM-L0	1.06 ± 0.01	3.41 ± 0.04	7.30 ± 0.04	1.83 ± 0.05
	JM-L0.1	1.14 ± 0.02	3.83 ± 0.06	8.02 ± 0.16	1.77 ± 0.05
	JM-L1	1.15 ± 0.01	3.99 ± 0.02	8.45 ± 0.17	1.83 ± 0.03
	JM-L10	1.08 ± 0.01	3.67 ± 0.03	7.25 ± 0.19	1.68 ± 0.04
Jet mill after HME	HME-L0	0.98 ± 0.01	3.23 ± 0.01	6.24 ± 0.04	1.63 ± 0.01
	HME-L0.1	1.07 ± 0.01	3.98 ± 0.11	8.09 ± 0.51	1.76 ± 0.08
	HME-L1	1.05 ± 0.01	3.94 ± 0.02	8.34 ± 0.09	1.85 ± 0.03
	HME-L10	0.97 ± 0.01	3.21 ± 0.03	7.44 ± 0.19	2.02 ± 0.06

**Table 3 pharmaceutics-16-00784-t003:** Aerodynamic performance characteristics of JMs and HMEs, including ED, FPF, MMAD, and GSD (mean ± SD, n = 3).

Formulation		ED (%)	FPF (%) <4.46 μm	eFPF (%) <1.66 μm	MMAD (μm)	GSD
Co-jet mill	JM-L0	77.06 ± 2.93	69.50 ± 4.17	22.61 ± 1.13	3.91 ± 0.06	1.50 ± 0.66
	JM-L0.1	85.40 ± 1.25	58.72 ± 4.42	15.77 ± 0.79	4.50 ± 0.12	1.40 ± 0.58
	JM-L1	90.19 ± 2.00	61.64 ± 1.44	13.88 ± 0.82	5.10 ± 0.13	1.44 ± 0.59
	JM-L10	93.68 ± 0.46	70.57 ± 0.35	16.92 ± 0.92	4.74 ± 0.14	1.42 ± 0.57
HME + jet mill	HME-L0	84.95 ± 2.19 ^###^	65.83 ± 4.36	23.14 ± 2.13	3.71 ± 0.14	1.94 ± 0.01
	HME-L0.1	91.45 ± 1.66 ^#^	55.29 ± 0.62	19.54 ± 4.33	4.68 ± 0.43	1.34 ± 0.92
	HME-L1	93.39 ± 0.39	61.57 ± 4.02	21.34 ± 2.94 *	4.42 ± 0.31	2.08 ± 0.39
	HME-L10	94.18 ± 0.74	77.53 ± 0.87	33.68 ± 1.31 ****	3.46 ± 0.06	2.11 ± 1.31

^###^ ANOVA, *p*-value < 0.001 compared with JM-L0; ^#^ ANOVA, *p*-value < 0.05, compared with JM-L0.1; * ANOVA, *p*-value < 0.05, compared with JM-L1; **** ANOVA, *p*-value < 0.0001 compared with JM-L10.

## Data Availability

The datasets for this work are available from the authors upon request.

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
