# Peer review of "Preparation and Evaluation of Inhalable Microparticles with Improved Aerodynamic Performance and Dispersibility Using L-Leucine and Hot-Melt Extrusion"

_pharmaceutics, 2024, doi:10.3390/pharmaceutics16060784_

Round 1

Reviewer 1 Report

Comments and Suggestions for Authors

This article reports that application of hot-melt extrusion to dry powder inhaled formulation to overcome issue of spray-dried particle. The knowledge shown in the manuscript is worthy for publication of Pharmaceutics following minor revisions. The comments are listed here and the authors should consider them.

1.     Space between number and symbol may often be typo through the manuscript. For example, 33.68±1.31% (Abstract), 0.5mm to 5 mm (page 2, line 51), and 160 °C (page 3, line 119). They should be written as 33.68 ± 1.31%, 0.5 mm to 5 mm, and 160°C? Please confirm the author instruction and other articles in Pharmaceutics.

2.     In the method, mixed LEU was written as a ratio of 10 (page 3, line 104 and 116). However, Table 1 shows the experimental ratios of LEU at 0, 0.1, 1, and 10. Please revise them to correct values.

3.     The temperature HME performed was set as 160°C based on the melting point of ITZ (page 3, line 119). However, the melting point of ITZ was around 170°C (Figure 2). Please add further explanation of reason for setting temperature below the melting point.  If the author set -10°C from the melting point, please mention so with citation of references.

4.     “2.5. In-vitro aerodynamic performance study” (page 5, line 198) is typo? This title was used in “2.4” and the contents of 2.5. were physicochemical properties.

5.     Dv90 range is written as 6 to 8 mm (page 7, line 304). However, the range of Dv90 shown in Table 2 is 6.24 to 8.45 mm, which was not in 6-8 mm. Please re-write the correct values.

6.     Decrease in the crystallinity of ITZ after HME was reported (page 9, line 358). However, the XRPD profiles of HME seems similar to that of JM (Figure 3D and C). Please confirm them, Figure 3D might is same to Figure 3C.

7.     “Leu” is “LEU”? (page 16, line 522).

8.     Change in solubility, water sorption, and contact angle of HME as compared to JM is shown in Figure 6-8. However, the authors proposed flowability is an issue for spray-dried DPI (page 2, line 52). Hence, to overcome this issue is essential for this investigation. Why not measure flowability of HME samples and show the result is better as pharmaceutical formulation.

Author Response

Please see the attachment. Please review the changes by selecting "Review - Tracking - All Markup" or "No Markup" in Word to see all comments and modifications.

Reviewer 2 Report

Comments and Suggestions for Authors

 The manuscript "Preparation and evaluation of inhalable microparticles with improved aerodynamic performance and dispersibility using L-leucine and hot-melt extrusion" explores the enhancement of dry powder inhalers (DPIs) by incorporating L-leucine (LEU) using hot-melt extrusion (HME). The study demonstrates that LEU significantly improves the dispersibility and aerodynamic performance of itraconazole (ITZ) microparticles. Key findings include increased fine particle fraction (FPF) and optimal mass median aerodynamic diameter (MMAD), validated through various characterization techniques. I believe the work should be ready for publication after minor revision.

Introduction

I believe the introduction would benefit from an explanation of previous studies dealing with these issues. What has been done before to solve this?

Materials and Methods

Line 108: Please provide some context for the jet milling conditions, why this pressure?

Line 119: Is the temperature the same throughout the extruder?

Line 132: any pre-treatment before measurement?

Line 157: how did you prepare the powder for characterization?

Line 296: Please add a statistical analysis subsection.

Results and discussion

Line 395: “not result in significant differences” is this comment verified by a statistical treatment? What p value was used? If not, please change the word “significant”.

Line 338: please provide a comparison between HME particles 0.1 to 1.0 to 10

Line 350: Wouldn’t it be preferable to have ITZ amorphous as well? From this HME did not produce amorphous ITZ. Please comment.

Line 355: What PM particles refer to? Is this the powder mixed after the turbula? 

Line 367: can you calculate the crystallinity of the mix?

Line 385: it would be helpful to have a brief explanation of how ED, FPF and EFPF are related to the drug delivery to the lungs.

Line 408: please discuss how particle properties influence deposition at each NGI stage and this correlates with different lung regions

Line 482: this solubility section basically describes the results and provides no discussion. Provide a more in-depth discussion of the mechanisms behind the enhanced solubility observed in HME formulations, particularly focusing on the role of LEU and the HME process

Line 518: please provide some discussion regarding the DVS findings and storage and handling of DPI formulations.

Line 575:  please discuss briefly limitations of the study for instance potential variability in batch-to-batch consistency

Author Response

(The authors gave the same response as above.)

Reviewer 3 Report

Comments and Suggestions for Authors

It is an interesting study that preparation and evaluation of inhalable microparticles with improved aerodynamic performance and dispersibility using L- leucine and hot-melt extrusion. The reviewer suggests a Minor Revision and there are some suggestions for authors:

1.     Why choose L-leucine as an excipient in inhalable dry powders? What are the advantages compared with other Amino acids?

2.     How does L-leucine concentrates on surface of the spray droplets? Please add more details.

3.     Please add specific melting point information for ITZ and L-leucine.

4.     In the preparation of itraconazole, why is the ratio of ITZ to MAN 2:8? And not some other ratio of 3:7 or 5:5?

5.     Please add a figure to illustrate how the ITZ micropaticles were prepared by HME.

Author Response

(The authors gave the same response as above.)
